# Enhancing Resveratrol Bioproduction and Anti-Melanogenic Activities through Elicitation in DJ526 Cell Suspension

**DOI:** 10.3390/plants10081653

**Published:** 2021-08-11

**Authors:** Vipada Kantayos, Jin-Suk Kim, So-Hyeon Baek

**Affiliations:** Department of Agricultural Life Science, Sunchon National University, Suncheon 57922, Korea; vkwah@naver.com (V.K.); kimjs6911@naver.com (J.-S.K.)

**Keywords:** rice, elicitors, resveratrol, cell suspension, gene expression, tyrosinase

## Abstract

Resveratrol, a secondary plant metabolite, and its derivatives, including piceid, show several potential health-related biological activities. However, resveratrol production is uncommon in plants; thus, resveratrol-enriched rice (DJ526) is produced for its nutritional and therapeutic value. Here, a DJ526 cell suspension was treated with various elicitors to determine its resveratrol-production potential and elicit its biological activity. Treatments with most elicitors produced more piceid than resveratrol; as elicitation periods increased, the average piceid levels were 75-fold higher than resveratrol levels. This increase is associated with glycosylation during growth and development. The duration of exposure and concentrations of elicitors were crucial factors affecting resveratrol synthase expression. Of all the elicitors tested, jasmonic acid and methyl jasmonate (MeJA) were strong elicitors; they increased resveratrol production to ≤115.1 μg g^−1^ (total resveratrol and piceid content). Moreover, 5 μM of MeJA increased total resveratrol production by >96.4% relative to the control production. In addition, the extract of cell suspension treated with 5 μM of MeJA significantly reduced melanin content and cellular tyrosinase activity (24.2% and 21.5% relative to the control, respectively) in melan-a cells without disturbing cell viability. Overall, elicitation can enhance resveratrol production and elicit the biological activity of the compound, in this case, its anti-melanogenic activities, in DJ526 cell suspension.

## 1. Introduction

Resveratrol (3,5,4’-trihydroxy-trans-stilbene) is a secondary metabolite of the stilbene group that is mostly found in grapes, peanuts, and berry fruits [1]. Over many years, the biological activities and therapeutic potentials of resveratrol have been reported, with cardioprotective and chemoprotective effects among its potential health benefits [2]. Because resveratrol is an uncommon representative compound in plants, resveratrol-enriched rice (DJ526) was developed using genetic modification technology to increase the nutritional value and therapeutic utility of Korean rice [3]. In DJ526, resveratrol is produced in all tissues, but particularly in the seeds and callus, via expression of the resveratrol synthase (*RS*) gene of the peanut Palkwang variety. The seeds and callus of DJ526 were shown to possess many potential bioactivities such as caloric restriction [4], physical strength improvement [5], and lifespan extension in fruit flies [6].

However, the presence of resveratrol (aglycone) can be limited by the glycosylation process, which is important for plant growth, development, and defensive responses [7,8], and its key enzyme glycosyltransferase. Glycosylation modifies the structures of secondary metabolites by changing aglycone to a glycoside form. Resveratrol has a large number of glycoside derivatives derived from enzymatic reactions through biosynthetic pathways [9]. One major resveratrol derivative is glycoside piceid (polydatin), which contains one sugar molecule at the C-3 position. Both resveratrol and piceid exhibit in vitro and in vivo health-related bioactivities, e.g., anti-oxidant activity, anti-cancer activity, and cardiovascular disease prevention [10,11,12,13].

The productivity of secondary metabolites has an impact on their biological activities. To improve the production of bioactive compounds in plants, a variety of plant secondary metabolites can be elicited. Elicitation is a process wherein an elicitor stimulates the production of secondary metabolites or bioactive substances through plant defense mechanisms. There are two major groups of elicitors: biotic and abiotic. Biotic elicitors are obtained from biological sources, such as plants, animal structures, and microorganisms (e.g., chitosan (CHI), pectin, cellulose, yeast extract (YE), and fungal extract), whereas abiotic elicitors are chemical substances, such as signaling molecules, salicylic acid, jasmonic acid (JA), methyl jasmonate (methyl ester of JA (MeJA)), and heavy metals, as well as physical stimuli including UV radiation [14,15]. Elicitors such as JA, salicylic acid, CHI, UV, and some plant mineral nutrients, e.g., nitrogen and potassium, have the capacity to improve resveratrol production [16]. To promote DJ526 as a biomaterial source, it can be cultivated on a commercial scale through cell suspension cultures, given the potential quality, consistency, and time-efficiency of this method when applied to large-scale production.

In this study, DJ526 cell suspensions were subjected to various elicitor conditions, i.e., treatment with YE, CHI, UV, sonication, JA, and MeJA at various concentrations over various elicitation periods. The aim was to identify the ideal elicitor with a favorable concentration, specifically one that showed the highest resveratrol bioproduction (compared with the control) that could be chosen to evaluate the relationship between glycosylation and the quantity of resveratrol bioproduct. In vitro glycosylation during cell suspension culture is related to the change in secondary metabolite production [17]; therefore, changes in resveratrol and piceid production during elicitation were determined via gene expression analysis of candidate glycosyltransferases. Additionally, the *RS* gene activity of elicitor-treated DJ526 cell suspensions was investigated. The extract of elicitor-treated DJ526 cell suspension was also tested for its anti-melanogenic activities (skin whitening properties) by assessing the cellular melanin content and anti-tyrosinase activity in melan-a cells. Overall, this paper provides a novel strategy for improving an active compound and its biological activity through the use of elicitation in resveratrol-enriched rice.

## 2. Results and Discussion

### 2.1. Potential Elicitors of Resveratrol and Piceid Content in DJ526 Cell Suspension

Analyses of resveratrol and piceid in DJ526 cell suspension were performed using an HPLC system, which showed the various resveratrol-production responses under different elicitors. The average resveratrol content in all treatments was ~75-fold lower than piceid content levels. Resveratrol and piceid in elicitor-treated DJ526 cell suspension were found at <5.0 and 16.0–87.5 μg g^−1^, respectively.

#### 2.1.1. Biotic Elicitors

In DJ526 cell suspension, the biotic elicitors YE and CHI acted as resveratrol product enhancers and inhibitors, respectively (Figure 1 and Figure 2). After YE treatment, piceid content was 25.4–80.4 µg g^−1^ (Figure 1a), whereas resveratrol content was lower at around 1.1–4.7 µg g^−1^ (Figure 1b). After YE treatment for 12 h, piceid content reduced by ~50%; it then increased almost two-fold after 24 h and decreased again after 48 h of treatment. In the case of resveratrol, the content was gradually increased for 24 h and decreased at 48 h. The highest volumes of piceid and resveratrol content were found after 24 h of YE treatments: 1.5- and 4.5-fold increases relative to the control levels, i.e., around 80.4 and 4.7 µg g^−1^, respectively (Figure 1). In explanation of this pattern, 12 h of YE treatment seems to have initially enhanced the stress response in cell suspension as piceid concentrations decreased, and then between 12 and 24 h, a critical elicitation period occurred during which cells might have recovered after stress memory was formed and signal molecules were released that enabled this recovery [18]. At 48 h, however, high levels of elicitation may have induced stress in culture conditions and showed a suppressive effect on piceid and resveratrol production.

With 150 μM CHI for 24 h, piceid and resveratrol contents were comparable; however, after 48 h, piceid content was lower than that of the control, whereas resveratrol content increased by 8% (Figure 2a,b). CHI reportedly had a positive effect on enhancing intracellular resveratrol yield in grape cell culture [19,20]; however, we found that CHI is ineffective for resveratrol production in DJ526 cell suspension.

#### 2.1.2. Abiotic Elicitors

After UV treatment at 312 nm (UV-B), resveratrol and piceid production significantly differed from that of the control. Additionally, piceid content decreased by 8.5–23.3% after exposure to UV at 365 nm (UV-A), whereas resveratrol was not detected (Figure 3a). Thus, UV-B had no effect on piceid and resveratrol content in DJ526 rice cell suspension, whereas UV-A reduced both piceid and resveratrol contents, particularly reducing piceid content as exposure time increased. It is possible that UV radiation overdose causes oxidative stress, leading to growth and development issues in the plant. With sonication treatment, the resveratrol and piceid contents did not differ from those in the control group (Figure 3b). In contrast, Potrebko and Resurreccion [21] found that UV and sonication treatments improved resveratrol concentrations by approximately 2 and 4 μg g^−1^, respectively, in treated peanut, indicating that UV and sonication may have beneficial effects.

Among the six elicitors, JA and MeJA elicitation most strongly stimulated resveratrol and piceid production, especially during 5 day elicitations. However, after 10 day treatments, piceid levels decreased and resveratrol was present either at trace levels or not at all (data not shown). The amounts of piceid and resveratrol produced after JA and MeJA treatments over 5 days were 19.4–111.8 and 1.9–3.9 µg g^−1^, respectively. The variation in piceid and resveratrol in both treatments was determined as a percentage of the control (Figure 4). MeJA at 5 µM produced the highest level of piceid (>99% that of the control), whereas the highest level of resveratrol (42.1%) was elicited by JA at 20 µM (Figure 4a,b). Thus, in a 5 day culture period, MeJA elicited more piceid and resveratrol production compared with JA. Taurino et al. [22] also reported that MeJA is more effective in eliciting intracellular resveratrol compared with JA. However, for potent elicitation of metabolite production, it is necessary to choose the optimum concentration, treatment duration, type of elicitor, and genotype of cell culture to achieve the best response against plant cell mechanisms [23]. The concentrations of resveratrol and stilbene derivatives are affected by several factors, including two major factors, i.e., genetic factor, environment factor (stress, climate condition), as Oh et al. [24] represented different concentration of resveratrol and stilbenes that were found from various food sources.

### 2.2. Elicitation Promotes Glycosyltransferase Activity

Changes in resveratrol and piceid content are the result of the glycosylation process in which glycosyltransferase transfers sugar residues to resveratrol aglycone to form glucoside piceid [25,26]. MeJA at a concentration of 5 µM was used as a model to illustrate the significant role of glycosyltransferase activity in the production of resveratrol and its derivatives, especially piceid (Figure 4a). Aside from an increase in piceid content after MeJA treatment, we also found an increase in an unknown compound (unk.Q) at the front position of the piceid peak (Figure 4c); this is probably related to piceid or is a resveratrol derivative compound as it was not detected in normal rice cell suspension culture (Figure 5). In a previous study by Khan et al. [13], resveratrol and piceid were not detected in normal rice (i.e., DJ rice). Therefore, the unknown peak was likely related to a resveratrol derivative compound. The expression of glycosyltransferase after 5 µM MeJA treatment was determined using real-time PCR with seven candidate glycosyltransferases selected from a previous study [27]. Upregulation of the putative glycosyltransferases in MeJA-treated DJ526 cell suspension was observed relative to the control cell suspension (Figure 6a,b). Among the seven candidates, 0.2–1.3-fold upregulation (fold change threshold ≥1) in the expression of glycosyltransferases, especially QR3, was the highest with MeJA treatment at 5 µM. All the candidate genes belonged to flavonoid glucosyltransferases and UDP glycosyltransferases.

### 2.3. Role of MeJA in RS Gene Expression Relative to Total Resveratrol Production

The total resveratrol production was considered as the total combined amount of resveratrol and piceid. Because DJ526 is transgenic rice with a modified *RS* gene, determining the expression of the *RS* gene in response to the elicitors is important for efficient resveratrol product enhancement. As our earlier findings indicated that JA and MeJA treatment elicited the highest piceid and resveratrol contents, these treatments were studied in relation to *RS* gene expression. Total resveratrol production from JA and MeJA ranged from 24.2 to 115.1 µg g^−1^. At 10 µM, JA elicited a 27.5% increase in total resveratrol production; at 5µM, MeJA elicited a 96.4% increase in total resveratrol production (both relative to the respective controls; Figure 7a). This percentage increase corresponded to a 0.5-fold (about 51.2%) increase in *RS* gene expression after 5 µM MeJA treatment, but the JA treatment produced no change in *RS* gene expression relative to that of the control (Figure 7b). JA and MeJA are plant signaling compounds that affect plant growth and gene expression [28]. Several studies have reported that MeJA is a potent elicitor of resveratrol production [29,30,31]. Ho et al. [32] reported that elicitation by MeJA begins in the plasma membrane, where the specific receptor receives the signal and switches on the defense mechanism in the plant; however, the signal transduction process elicited by MeJA is currently unclear.

### 2.4. Effect of DJ526 and MeJA-Treated Cell Suspension on Anti-Melanogenesis

Given the effects we reported thus far, 5 µM MeJA was used to examine its effects on anti-melanogenesis associated with melanin production and tyrosinase activity in melan-a cells. The results showed that the cell suspension treated with 5 µM MeJA extract (final concentration: 100 μg mL^−1^) reduced melanin production by 24.2% and 15.1% relative to the negative control and untreated cell suspension, respectively, without affecting cell viability (Figure 8a). In addition, intracellular tyrosinase activity was inhibited by 21.5% and 11.0% compared with their activities in the negative control and untreated cell suspension, respectively. The potencies of the cell suspension treated with 5 µM MeJA extract and the positive control (arbutin) were indistinguishable in terms of their effect on melanin content and tyrosinase activity when the melan-a cells were treated with equivalent concentrations (Figure 8b). Furthermore, the expression of melanogenesis-associated genes, including *TRP-1*, *TRP-2*, *tyrosinase**,* and their transcription factor, *MITF*, were downregulated in MeJA-treated (5 µM) and untreated cell suspensions. In MeJA-treated cell suspension, 0.33-, 0.18-, and 0.2-fold downregulation of *MITF*, *TRP-1*, and *tyrosinase*, respectively, were observed, along with a 0.54-fold upregulation of *TRP-2* (Figure 8c). However, the expressions of *TRP-1* and *tyrosinase* did not change in the untreated cell suspension. These results show that extracts of MeJA-treated cell suspension and DJ526 cell suspension can inhibit melanogenesis at the transcription level through a reduction in *MITF* and *tyrosinase* expression. Importantly, the cell suspension elicited with MeJA seem to be better than the untreated cell suspension at inhibiting melanin production; thus, elicitation may be an effective approach for increasing the concentration of resveratrol and its derivatives, piceid, which may achieve higher biological activity [33].

## 3. Materials and Methods

### 3.1. Quantitation of Resveratrol and Piceid Content in Elicitor-Treated DJ526 Cell Suspension

#### 3.1.1. Callus Induction

DJ526 calli were induced from seeds on solid media containing 2 mg L^−1^ of 2,4-D [13]. After culturing for approximately 4 weeks, 5 g (fresh weight) of friable calli were suspended in liquid medium supplemented with 2 mg L^−1^ of 2,4-D before being incubated on a rotary shaker at 100 rpm under light conditions (200 μmol m^−2^ s^−1^) at 26 ± 1 °C. Cell suspensions were resuspended every 10 days with 200 mL of liquid medium in a 1000 mL Erlenmeyer flask to maintain fresh media conditions for the calli.

#### 3.1.2. In Vitro Elicitor Treatment Preparations

A forty-day-old cell suspension (10 g FW) was cultured in a 250 mL Erlenmeyer flask containing 50 mL of liquid culture containing the individual elicitors. YE (Becton, Dickenson and Company, Sparks, MD, USA) and CHI (Sigma, St. Louis, MO, USA) were used as representative biotic elicitor treatments, whereas UV radiation and sonication were performed as physical treatments, and JA (Sigma) and its methyl ester MeJA (Sigma) were used as abiotic elicitor treatments. These treatments were prepared as follows: 3 g L^−1^ of YE from autolyzed yeast cells was supplemented into the cell suspension culture with elicitation times of 12, 24, and 48 h; final concentrations of CHI at 100, 200, 500, and 1000 μg L^−1^ were added to cell suspension culture for elicitation times of 24 and 48 h; UV radiation and sonication were performed on the day of subculture, with transilluminators (DaiHan Scientific, Korea) used as the UV source (UV-A at 365 nm; UV-B at 312 nm) for 1, 3, 5, and 10 min and an ultrasonic bath (40 kHz) used for sonication (10, 20, and 30 min). JA and MeJA were prepared by dissolving them in absolute ethanol to a final stock solution of 5 mM followed by working concentrations of 0, 5, 10, 20, and 40 μM, which were applied to cell suspension for 5 and 10 days of elicitation. The cell suspension culture without the addition of elicitors was used as a control.

#### 3.1.3. Determination of Piceid and Resveratrol Content

Dried calli (0.1 g) were prepared in a hot-air oven at 60 °C for 2 days, and extracted using 80% methanol (1:10) and sonicated at 40 °C for 1 h. After centrifugation at 13,000 rpm, the supernatant was collected and filtered through a 0.45 µm syringe filter before being injected into the HPLC system (Waters e2695). The separated components were passed through a C18 reverse-phase column (4.6 × 150 mm; Waters). The mobile phase consisted of distilled water (A) and acetonitrile (B) with gradient elution as follows: 10% solvent A (37 min), 30% solvent A (7 min), and 10% solvent A (5 min). The flow rate was 1 mL/min and a UV-visible detector (Waters 2489) was applied at 308 nm. HPLC conditions were modified from the method of Baek et al. [3]. The LOD and LOQ of the standard mixture (piceid and resveratrol) were determined based on the standard deviation (SD) of the response and the slope of the calibration curve. The LOD was defined as the smallest amount of an analyte in a sample that could be detected, whereas the LOQ was considered the smallest amount of an analyte that could be determined with accuracy and precision [34,35]. The limit of detection (LOD) and limit of quantification (LOQ) of standard piceid are 0.50 and 1.52 μg g^−1^, respectively, whereas the LOD and LOQ of standard resveratrol are 0.26 and 0.79 μg g^−1^, respectively. The calibration curve presented good linear regressions with a correlation coefficient (R^2^) of >0.9997 and >0.9994 for piceid and resveratrol, respectively.

#### 3.1.4. Real-Time PCR for *RS* Gene and Glycosyltransferase Expression Analysis

The total RNA was extracted using the Tri-reagent (Invitrogen) and reverse-transcribed to cDNA using a Power cDNA Synthesis Kit (Intron Biotechnology) as per the manufacturer’s instructions. Quantitative PCR was performed using RealMOD^TM^ Green W^2^ 2× qPCR Mix (Intron Biotechnology). The *RS* gene and glycosyltransferase expression were analyzed using a real-time PCR detection system (Biorad, CFX96 connect real-time PCR). The primers used for studying glycosyltransferase expression were selected from a previous study (Table 1) [3,27]. The PCR conditions were as follows: 95 °C for 10 min, followed by 40 cycles at 95 °C for 15 s and 60 °C for 30 s (two-step). Data are expressed as relative normalized gene expressions. Actin was used as the internal control.

### 3.2. In Vitro Skin Cell Culture

#### 3.2.1. Cell Culture and Viability

Melan-a cells were cultured in RPMI1640 media containing 10% fetal bovine serum, penicillin (50 U mL^−1^), streptomycin (50 µg mL^−1^), and TPA (200 nM). Cells were maintained at 37 °C with 5% CO_2_ until the determination of anti-melanogenesis [36]. The viability of melan-a cells was determined by the cleavage of water-soluble tetrazolium salt to formazan dye, which relies on the activity of dehydrogenase in the mitochondria of living cells, using an Ez-cytox Assay Kit (DogenBio, Seoul, Korea). The melan-a cells (2 × 10^5^) were seeded onto 96-well plates and cultured overnight at 37 °C with 5% CO_2_, after which the cells were treated with elicitors and brought to a final concentration of 100 μg mL^−1^. DMSO was used as a negative control, untreated DJ526 cell suspension extract as a treatment control, and arbutin as a positive control. Reagent was added to each well and incubated at 37 °C for 4 h, after which the absorbance signal was measured at 450 nm.

#### 3.2.2. Determination of Melanin Content and Tyrosinase Activity

The melanin content and tyrosinase activity were measured using the protocol of Seo et al. [36] with some modifications. Briefly, melan-a cells (8 × 10^4^) were seeded onto 6-well plates and cultured overnight before treatment with 100 μg mL^−1^ of cell suspension extracts. After a 3 day incubation with elicitor-treated cell suspension extracts (100 μg mL^−1^), the melan-a cells were collected by trypsinization and then centrifuged at 2000 rpm for 10 min. The obtained pellets were first lysed using Triton lysis buffer supplemented with 1% PMSF and then placed on ice for 30 min; after this, the lysates were centrifuged at 13,000 rpm. The supernatant was used to determine tyrosinase activity, whereas the pellets were used to evaluate cellular melanin content. For tyrosinase activity, the protein level was measured using the Bradford protein assay. The protein (40 μg) was adjusted with a lysis buffer and then incubated with L-dopa (2 mg mL^−1^) at 37 °C for 1 h. Pellets were dissolved in 1 N NaOH (10% DMSO) and incubated at 80 °C for 1 h. Cellular tyrosinase activity and melanin content were measured at 405 nm.

#### 3.2.3. Melanin Synthesis Gene Expression Using Real-Time PCR Analysis

Melan-a cells (8 × 10^4^) were seeded onto 6-well plates and were treated with cell suspension extracts and cultured for 3 days (as in our previously described experiment). Cells were lysed using the Tri-reagent to extract the total RNA and then cDNA was synthesized according to the manufacturer’s instructions (Intron Biotechnology). Real-time PCR was performed as described in Section 3.1. The following melanogenesis-specific primers were designed with sense and antisense nucleotides; *MITF* (5′-TGAAGGTCGGTGTGAACGGATTTCGC-3′ and 5′-CATGTAGGCCATGAGGTCCACCAC-3′), *TRP-1* (5′-ACTTCACTCAAGCCAACTGC-3′ and 5′-AGCTTCCCATCAGATGTCGT-3′), *TRP-2* (5′-GCTCCAAGTGGCTGTAGACC-3′ and 5′-AATGCAGTGGCTTGGAAATC-3′), and *tyrosinase* (5′-GACGGTCACTGCAGACTTTG-3′ and 5′-GCCATGACCAGGATGAC-3′). *GAPDH* (5′-AACTTTGGCATTGTGGAAGG-3′ and 5′-ACACATTGGGGGTAGGAACA) was used as an internal control.

### 3.3. Statistical Analysis

Data are presented as the means ± SD of three independent replications. Statistical differences were determined by one-way ANOVA using MS Excel 2019. A *p*-value of <0.05 was considered statistically significant.

## 4. Conclusions

In the current study, we demonstrated an essential elicitation process for resveratrol production in DJ526 cell suspension by the high productivity of piceid and resveratrol. We found that the concentration of piceid was consistently higher than that of resveratrol due to glycosylation during growth and development. Glycosylation plays a role in secondary metabolite production as well as in their bioavailability and biological activity. We found that JA and MeJA are potent elicitors that enhanced resveratrol production in DJ526 cell suspension. The elicitation-mediate MeJA induced glycosyltransferase activity, as shown by the upregulation of candidate resveratrol-dependent glycosyltransferases, which belong to flavonoid glucosyltransferases and UDP-glycosyltransferases. Finally, the DJ526 cell suspension extract treated with 5 μM MeJA produced anti-melanogenic effects in melan-a cells by suppressing tyrosinase activity and melanin pigment in these cells.

## Figures and Tables

**Figure 1 plants-10-01653-f001:**
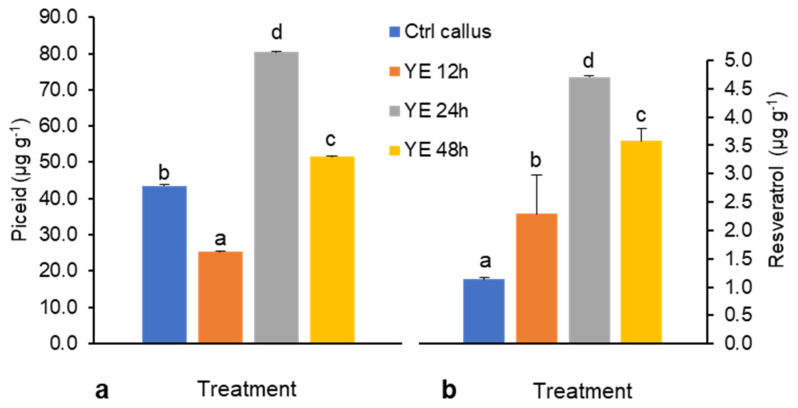
(**a**) Piceid and (**b**) resveratrol content elicited by 3 g L^−1^ yeast extract (YE) treatment for 12, 24, and 48 h. Yield of piceid and resveratrol are presented as means ± SD of three experiments. Means were compared with Duncan’s multiple range test (DMRT) at *p* < 0.05. Different letters (a–d) indicate significantly difference.

**Figure 2 plants-10-01653-f002:**
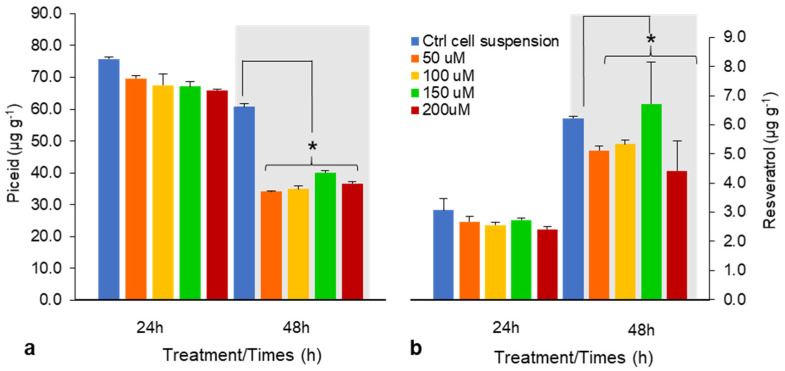
Effects of chitosan (CHI) on piceid (**a**) and resveratrol content (**b**) after 24 and 48 h treatments (50–200 μM). Piceid and resveratrol yields are presented as means ± SD of three experiments. * *p* < 0.05 compared with untreated DJ526 cell suspension (control).

**Figure 3 plants-10-01653-f003:**
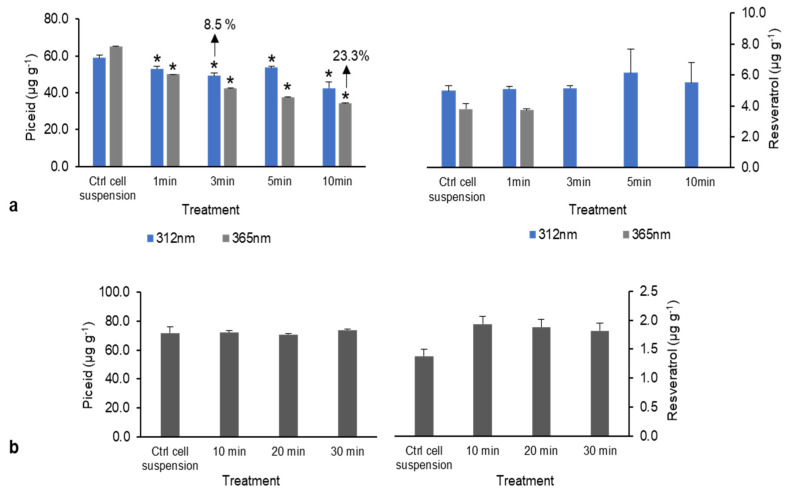
Piceid and resveratrol content following physical treatment. (**a**) UV radiation (UV-B at 312 nm and UV-A at 365 nm). (**b**) Sonication (10, 20, and 30 min). * Denotes significant differences at *p* < 0.05) between treatment and control groups.

**Figure 4 plants-10-01653-f004:**
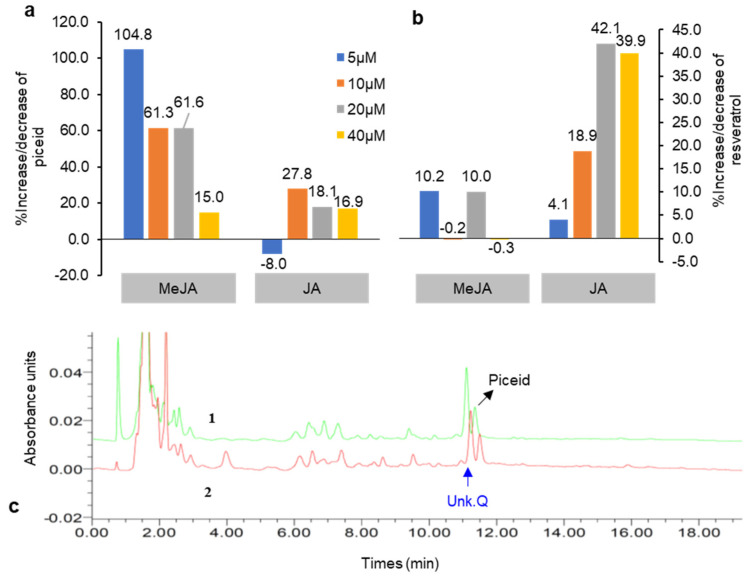
Percentage increase and decrease in piceid (**a**) and resveratrol content (**b**) in DJ526 cell suspension after treatment with jasmonic acid (JA) and methyl jasmonate (MeJA) under 5-day elicitation periods (compared with the untreated control). (**c**) Chromatogram representing an increase in piceid and unknown compound Q content in the 5 day MeJA (5 μM)-treated cell suspension (**1**) relative to the control (**2**).

**Figure 5 plants-10-01653-f005:**
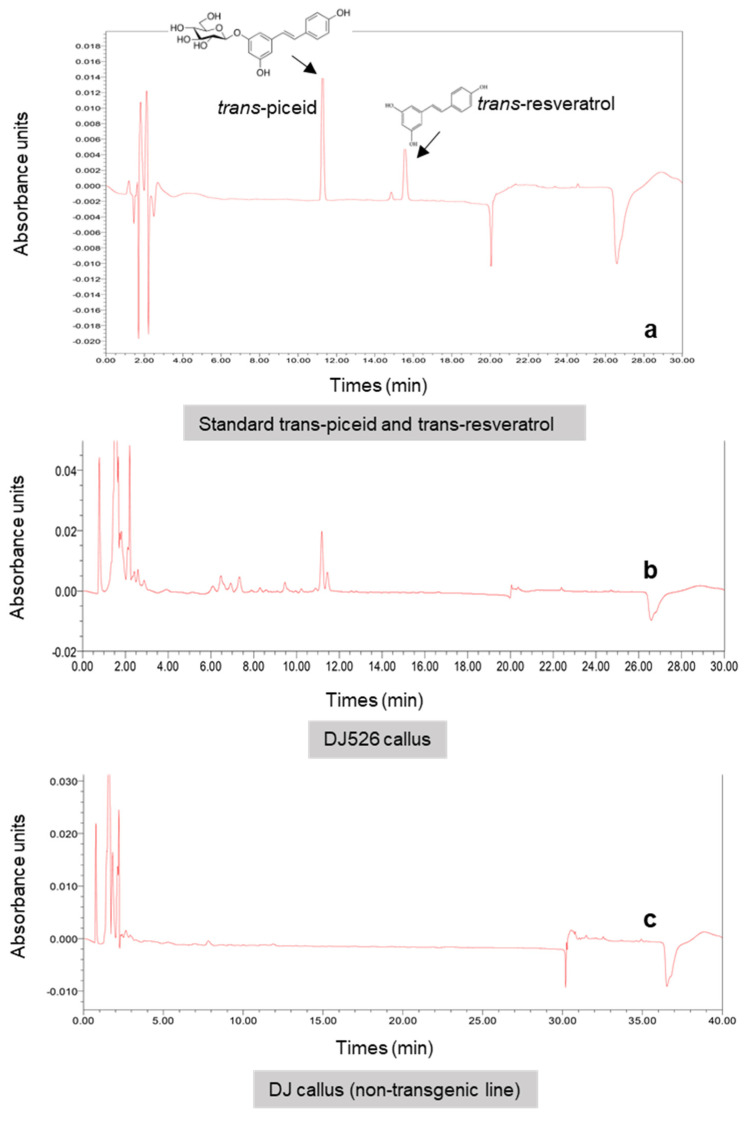
Chromatograms showing the detection of piceid and resveratrol in methanolic extracts of standard (**a**) and DJ526 rice (**b**), as well as the lack in non-transgenic rice (DJ) (**c**).

**Figure 6 plants-10-01653-f006:**
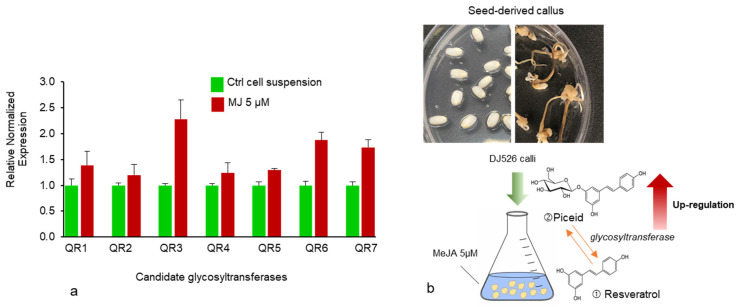
(**a**) Relative normalized expression of candidate UDP-glycosyltransferase genes after 5 μM MeJA treatment compared with the control group. Expressions are presented as means ± SEM. (**b**) Schematic representation of in vitro DJ526 cell suspension culture. Calli were induced on 2,4-D supplement media and the suspension culture was treated with 5 µM MeJA, which led to an increase in glycosyltransferase activity.

**Figure 7 plants-10-01653-f007:**
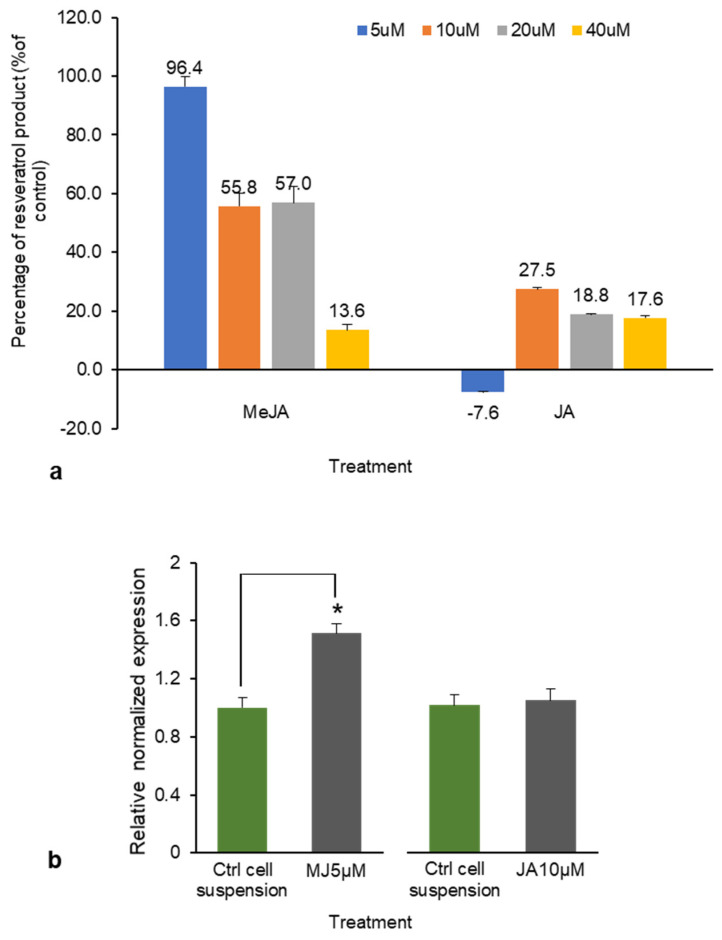
(**a**) The percentage increase/decrease in total resveratrol production (piceid and resveratrol combined) with methyl jasmonate (MeJA) and jasmonic acid (JA) treatments. (**b**) *RS* gene expression of cell suspension treated with MeJA (5 μM) and JA (10 μM) relative to the expression in the control arbitrarily defined as 1. * *p* < 0.05.

**Figure 8 plants-10-01653-f008:**
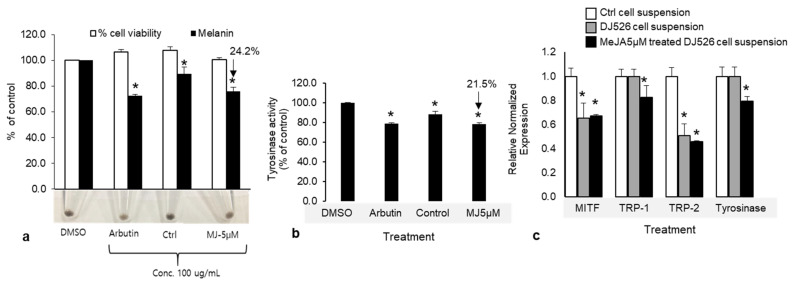
(**a**) Effects of methyl jasmonate (MeJA)-treated cell suspension extracts on melanin production and viability in melan-a cells. Cells (8 × 10^4^) were treated with 100 mg mL^−1^ of cell suspension extract and arbutin (positive control) for 3 days. (**b**) Intracellular tyrosinase activity of 5 μM MeJA-treated and untreated cell suspension. Cell viability, melanin content, and tyrosinase activity are presented as percentages of the control (DMSO). Arbutin was used as a positive control. Data are presented as means ± SD of three independent experiments. * *p* < 0.05. (**c**) The expressions of melanogenic related genes, *MITF*, *TRP-1*, *TRP-2*, and *tyrosinase*, in MeJA-treated cell suspension and DJ526 cell suspension (treatment control).

**Table 1 plants-10-01653-t001:** Seven candidate resveratrol-dependent glycosyltransferases.

No.	Gene	Gene ID	Description
1	RS	AF227963	stilbene synthase 3-like [Arachis hypogaea (peanut)]
2	QR1	LOC4341323	UDP-flavonoid 7-O-glucosyltransferase 73C6 (all organs) UDP-glycosyltransferase TURAN
3	QR2	LOC4326220	UDP-flavonoid 7-O-glucosyltransferase 73C6 (all organs)
4	QR3	LOC4335169	UDP-glycosyltransferase 74F2 (all organs)
5	QR4	LOC43335168	UDP-glycosyltransferase 74F2 (seeds)
6	QR5	LOC4335166	UDP-glycosyltransferase 74E2 (all organs) indole-3-acetate beta-glucosyltransferase
7	QR6	LOC4334167	UDP-glycosyltransferase 83A1 (all organs)
8	QR7	LOC4333838	UDP-glycosyltransferase 91B1

## Data Availability

The data presented in this study are available within this article.

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
