# Peer review of "Enhancing Resveratrol Bioproduction and Anti-Melanogenic Activities through Elicitation in DJ526 Cell Suspension"

_plants, 2021, doi:10.3390/plants10081653_

Round 1

Reviewer 1 Report

Dear all,

below are my comments and suggestions:

The article is well written and focused, and conforms to the aims and scope of the journal, so I suggest minor revision of this manuscript.

Please check your English (errors in spelling, grammar, and style) with a native English speaker.

Please indicate the bibliographic sources for Section 3.1. Quantitation of resveratrol and piceid content in elicitor-treated DJ526 callus

Please verify the number section of line 248

Add the bibliographic sources for Section 3.2. In vitro skin cell culture

Please verify the number section of line 281

The number of bibliographic sources is adequate, than 50% of the total bibliographic sources are from the last 5 years.

Author Response

Dear Reviewer1

Reviewer 2 Report

The manuscript entitled “Enhancing Resveratrol Bioproduction and Anti-melanogenic Activities by Elicitation in DJ526 Rice Callus” conforms to requirements of the journal “Plants” that are defined in the instruction for authors. The manuscript is very well written; the topic is interesting not only from scientific point of view, but also for its practical value. The study deals with the enhancement of Resveratrol production in DJ526 Rice cell suspensions. Resveratrol is a secondary metabolite exhibits various effects – antioxidant, anticancer, antifungal and anti-aging activity and possesses a number of human health benefits.

The abstract is enough informative and gives a brief account of the objective and the significance of the subject:  DJ526 callus was treated with various elicitors with the aim of determining their resveratrol-production potential and elicitation of biological activity. The relationship between glycosylation and the quantity of resveratrol bioproduct was also evaluated. In addition, elicitor treated DJ526 callus was tested for skin whitening properties by assessing cellular melanin content and anti-tyrosinase activity in melan-a cells.

The experimental results are arranged in four sub-sections, to present: A potentiality of elicitors in resveratrol and piceid content in DJ526 callus; Elicitation promotes glycosyltransferase activity; Role of MeJA in RS gene expression relative to total resveratrol production; Effect of DJ526 callus and MeJA-treated callus on antimelanogenesis.

The obtained results are successfully and adequately interpreted in the discussion, and their conclusions are logical. Authors compare the experimental results with those of other authors related to the subject.

After Minor Revision I suggest, the manuscript can be accepted in Plants. 

Specific comments 

Please, find my suggestions:

In Introduction section:

Page 2  from line 71  to line 76 “We found that JA and MeJA were ……….. ”should be moved to Results and Discussion section

In Results and Discussion:

Page 6 line 164  especially R9  should be corrected to especially R3  

In Materials and Methods:

Page 10, line 260 The primers used for studying RS gene expression were not described.

Author Response

Dear Reviewer2

Reviewer 3 Report

The manuscript brings about an interesting topic but the authors must address some points to improve the quality of the manuscript. 

“This process functions in the modification of secondary metabolite structures by changing aglycone to a nontoxic form (glycoside).”

I understand that the authors suggest that the aglycone form is toxic. Please clarify, toxic to what and under which condition?

The authors should clarify the novelty of the study at the end of the introduction.

Section 2.1, the authors should compare the concentration of resveratrol and piceid considering literature data for other sources. For that the authors can used the following paper.

Oh, W. Y., Gao, Y., & Shahidi, F. (2021). Stilbenoids: chemistry, occurrence, bioavailability and health effects—a review. Journal of Food Bioactives13. https://doi.org/10.31665/JFB.2020.13256

Fig. 6, statistical treatment is missing.

Fig. 7A, statistical treatment is missing

“elicitation may be an effective approach for improving the biological activity of secondary metabolites including resveratrol and piceid.”

This is not was taking place. The authors could state that “Elicitation may be an effective approach for increasing the concentration of resveratrol and its derivative, piceid, which may render a higher biological activity.” A good example of modification/improvement of biological activity may be found in the following paper.

Oh, W. Y., Chiou, Y.-S., Pan, M.-H., & Shahidi, F. (2019). Lipophilised resveratrol affects the generation of reactive nitrogen species in murine macrophages and cell viability of human cancer cell lines. Journal of Food Bioactives7. https://doi.org/10.31665/JFB.2019.7201

The authors could use the opportunity to mention this paper as it also shows the anti-inflammatory potential of resveratrol, which was not mentioned by the authors.

The authors must clarify how much was the inhibition of melanin production of extracts obtained from untreated and treated callus (5-μM methyl jasmonate (MeJA)-treated callus). Is the difference between the values significant?

The authors should also explain the relevance of using melan-a cells. What would be the expected biological importance of their findings.

Section 1.1.3, the authors should describe the HPLC system. The authors should also provide the limit of detection, limit of quantification, and R2 for the standard curve.

It is important to mention that all references mentioned here were purely based on COPE Ethical Guidelines for Peer Reviewers (http://publicationethics.org/files/Ethical_guidelines_for_peer_reviewers_0.pdf) that states that “suggestions must be based on valid academic or technological reasons.”

For example, Dr. Fereidoon Shahidi, the senior author of some manuscripts suggested here, is well known in the field of functional foods. In fact, according to google scholar, he is the number one in citations in the field of functional foods.

https://scholar.google.com/citations?view_op=search_authors&hl=en&mauthors=label:functional_foods

 Therefore, suggesting his literature is self-explanatory.

Author Response

Dear Reviewer3

Round 2

Reviewer 3 Report

The authors used a non-scientific source to answer the following question made in the first round of review:

  1. “This process functions in the modification of secondary metabolite structures by changing aglycone to a nontoxic form (glycoside).”

I understand that the authors suggest that the aglycone form is toxic. Please clarify, toxic to what and under which condition?

This was the answer:

Aglycones, when hydrolyzed (chemically degraded by the introduction of water molecules between adjacent subunits), are toxic to plants. The toxicity resides in the aglycone component or part of it. Thus, in plants, most of the compounds that are unstable, toxic, and hydrophobic or volatile occur in glycosylated forms. 

I found almost the same information/text in this website

https://www.britannica.com/science/aglycone

In addition, I recommended to provide the limit of detection, limit of quantification, and R2 for the standard curve. This was their answer:

Response: We have added these parameters to the revised manuscript: Results (page 3, line 95) and Materials and Methods (section 3.1.3, page 13, line 903).

However, the numbers representing the limit of detection, limit of quantification, and R2 were not provided.
